# Simulating and Verifying a 2D/3D Laser Line Sensor Measurement Algorithm on CAD Models and Real Objects

**DOI:** 10.3390/s24227396

**Published:** 2024-11-20

**Authors:** Rok Belšak, Janez Gotlih, Timi Karner

**Affiliations:** Laboratory for Robotisation, Faculty of Mechanical Engineering, University of Maribor, 2000 Maribor, Slovenia; rok.belsak@um.si (R.B.); janez.gotlih@um.si (J.G.)

**Keywords:** 2D/3D laser line sensor, profilometry, simulation, point cloud, measurement generation, STL, Matlab

## Abstract

The increasing adoption of 2D/3D laser line sensors in industrial and research applications necessitates accurate and efficient simulation tools for tasks such as surface inspection, dimensional verification, and quality control. This paper presents a novel algorithm developed in MATLAB for simulating the measurements of any 2D/3D laser line sensor on STL CAD models. The algorithm uses a modified fast-ray triangular intersection method, addressing challenges such as overlapping triangles in assembly models and incorporating sensor resolution to ensure realistic simulations. Quantitative analysis shows a significant reduction in computation time, enhancing the practical utility of the algorithm. The simulation results exhibit a mean deviation of 0.42 mm when compared to real-world measurements. Notably, the algorithm effectively handles complex geometric features, such as holes and grooves, and offers flexibility in generating point cloud data in both local and global coordinate systems. This work not only reduces the need for physical prototyping, thereby contributing to sustainability, but also supports AI training by generating accurate synthetic data. Future work should aim to further optimize the simulation speed and explore noise modeling to enhance the realism of simulated measurements.

## 1. Introduction

Two/three-dimensional laser line sensors have become vital tools in both industrial and research environments, particularly in applications where high-precision measurement is required. These sensors are utilized for tasks such as surface inspection, dimensional verification, and quality control in production lines, where determining whether a product meets specific dimensional tolerances is critical. Their versatility and high accuracy make them integral to advanced manufacturing processes and automation [1]. The increasing range of laser line sensors, combined with varying levels of accuracy, enables their application across different industries, from automotive to aerospace, and even in sectors such as heritage preservation [2]. The development of simulators for laser systems has been a growing field. Al-Temeemy [3] developed a 3D LADAR simulator that focuses on simulating the performance of laser detection and ranging (LADAR) systems using a fast target impulse response generation approach. The simulator models the laser’s beam propagation and interaction with the environment, providing accurate simulations of noise, resolution, and beam width. This approach is particularly beneficial for simulating systems in a wide variety of conditions and offers enhanced computational speed compared to traditional methods. In line with the need for fast 3D data processing, An et al. [4] presented a plane extraction method for 3D environments using a nodding 2D laser scanner. The method focuses on simplifying raw 3D data by extracting line segments to represent planes, significantly reducing computation time while maintaining data accuracy. Their research demonstrated that this approach is three times faster than existing methods. Antón et al. [2] highlighted the importance of 3D laser scanning in heritage preservation, specifically focusing on generating 3D CAD models from point clouds. Their work emphasizes the need for accurate as-built models to support future conservation efforts. The study illustrates how point cloud data can be transformed into detailed CAD models for heritage buildings, ensuring accurate monitoring and conservation of historical structures. Schlarp et al. [5] introduced an innovative approach to optical scanning for 3D imaging using laser triangulation sensors. The study aimed to optimize scanning performance by manipulating the optical path using tip–tilt mirrors, reducing the physical movement required by the sensor system. This approach demonstrated the potential for faster and more accurate scanning in various industrial applications. Additionally, Beermann et al. [6] explored the effects of inhomogeneous refractive index fields on laser triangulation measurements. Their work developed a simulation model that accounts for such variations, offering insights into improving the accuracy of laser-based measurements under different environmental conditions.

In the work of Mohammadikaji et al. [7], the simulation was carried out using the Mitsuba renderer, which handled the ray-tracing for coherent laser light and incoherent ambient light. The work simulates physically based light transport, simulating dominant optical effects and the spectral response of a particular digital imaging sensor. Two types of sensors were simulated: a color camera sensor and a monochromatic camera sensor, both utilizing EMVA 1288 specifications. The simulation described in the document used a cylinder head CAD model, and the entire process took approximately 10 h using eight CPU cores at 3.7 GHz.

In the work of Cajal et al. [8], laser triangulation sensors (LTSs) are simulated to optimize their design and ensure compliance with dimensional and geometrical tolerances in manufacturing processes. The simulation was performed in Visual C++ 2010 using the DirectX 9 SDK from Microsoft, specifically utilizing the Direct3D functions for simulation display and output generation. The simulation is tested using external CAD models of parts, allowing flexibility in the complexity and variety of the models used. The simulated scanning generates 2D coordinates of the laser on the surface and the 3D intersection of the laser plane with the part surface. While the exact time for the simulation is not specified, it is mentioned that the simulation environment reduces the prototyping time of real systems from several days to just minutes by simulating the spatial arrangement of the system components.

Roos-Hoefgeest et al. [9] addressed the challenge of simulating high-resolution profilometric sensors in the presence of noise. Laser profilometer measurements are simulated, specifically focusing on replicating the sensor readings in the presence of speckle noise using Perlin noise. The simulation is tested on various CAD models, including bearing caps, car doors, and heavy steel plates. The result of the simulated scan is a point cloud and 2D image where the size of the image is the total number of profiles multiplied by the number of points per profile. While the exact duration of the simulation is not explicitly mentioned, the goal is to efficiently replicate the scanning process and sensor noise, reducing time compared to physical testing. The simulation was implemented using C++ on the Ubuntu 20.04 LTS operating system, with the Visualization Toolkit (VTK) for handling 3D models and MATLAB 2022b for processing and analysis.

This research introduces a novel simulation algorithm for 2D/3D laser line sensors, specifically designed to handle complex STL CAD models, including assemblies with multiple overlapping parts. The research work is not focused on simulating the working principles of the laser line sensors, but it focuses on simulating measurement results of the 2D laser scans. None of the above-mentioned research has specified that they can scan assembly CAD models or have an option for generating point cloud data in different coordinate systems. By employing a modified fast-ray triangular intersection method, the algorithm enables efficient and accurate point cloud generation in both local and global coordinate systems. The method optimizes computation time while maintaining high fidelity to real-world measurements, reducing the mean deviation to just 0.42 mm This approach not only advances laser sensor simulation by integrating real sensor resolution but also offers a practical, sustainable alternative to physical prototyping. The algorithm’s ability to generate realistic synthetic data for AI training purposes further broadens its applicability across industrial and research domains.

Section 2 gives a detailed overview of the methods and principles that are used for simulation of measurements and resolution generation and describes a linear unit with a real sensor for verification of the simulated measurements. In Section 3, the results of the simulated measurements are given and evaluated in comparison to measurements from the real sensor. Section 4 provides a discussion about the methods used and evaluates the results between simulated and real measurements, while in Section 5, we draw conclusions and provide ideas for future work.

## 2. Materials and Methods

The main goal of the research is to generate measurements from CAD models which could be made up of only one part or of an assembly in STL format. The generated point cloud could be calculated in a laser coordinate system (LCS) and in a global coordinate system (GCS) since in some cases point cloud data are needed in a GCS. One example where point cloud data is needed in a GCS is for finding welding points of the weld joint for robotic welding [10]. In such cases, the trajectory of the laser regarding position and orientation is also needed to calculate the pose of the tool center point (TCP) of the robot to find the position and orientation of the welding joint. Resolution of the specific line laser could also be included in the measurements if needed. The simulation principle can be seen in Figure 1.

### 2.1. Generating Laser Lines

To simulate 2D/3D laser line sensor measurements, a specific laser needs to be considered. The 2D/3D MLWL 132 laser sensor from Wenglor company is simulated [11]. The sensor has the following characteristics shown in Table 1.

As can be seen, the working area of the selected laser has the shape of a trapezoid. To recreate the working area of the sensor and to obtain the direction of each laser line in one laser scan, vectors are being used, as can be seen in Figure 2. For each laser line, the start point, end point, and direction vector are calculated.

### 2.2. Modified Ray/Triangle Intersection Method

The proposed algorithm finds intersection points between a simulated laser line and an STL model consisting of triangles. The basis of the proposed algorithm is the fast, minimum storage ray/triangle intersection algorithm proposed by Möller and Trumbore [12]. The algorithm consists of a ray R defined by a vector origin O, a direction vector D, and a triangle consisting of three vertices, $V_{0},\;V_{1}\;$, and $V_{2}$, as can be seen in Figure 3. The equation of the ray is written as follows:(1)R(t)=O+t⋅D

A point, T(u,v), where u, v are barycentric coordinates, on a triangle is given by
(2)$$T(u,v)=(1-u-v)V_{0}+uV_{1}+vV_{2}$$
which must fulfill the following constraints, u≥0, v≥0 and u+v≤1, to lie inside the triangle.

The outlines of the ray/triangle intersection algorithm was found on the Matlab Exchange website written by Mena [13]. The algorithm finds the edges of the triangle first. Then, it computes the cross product of the direction vector D and one of the edges. The computed vector is perpendicular to the ray direction and one of the edges. The result is then multiplied as a dot product with the other edge to see if the result is close to zero. If it is zero or close to zero, it means that the ray and triangle are parallel and there is no intersection between them. If they are not parallel, the algorithm continues to compute the distance between the origin of the ray and the first vertices. If the barycentric coordinates are within the constraints that were defined before, it calculates the distance from the origin to the point on the triangle. The algorithm also returns the flag and coordinates of the barycentric points u, v.

However, during testing and investigating the code, some issues have been found while using the proposed algorithm for simulating measurement results of the 2D laser line sensor. The first observation was that if the triangle position was at the opposite side of the direction vector of the ray, the algorithm would still return the distance to the intersection point on the triangle. In this case, the distance was negative, so an additional condition was added that if the negative distance is calculated, the algorithm is terminated and zero is returned as a result. Another observation in the used algorithm was that the ray did not have any limitations on length. If the direction of the ray intersected with the triangle, the distance was calculated regardless of the ray length, which is not the case if one wants to simulate the measurements of the 2D laser line sensor with a limited working range, as can be seen in Figure 4.

The distance that should be considered while using the ray/triangle intersection algorithm is calculated from the working range of the sensor. Distance $L_{1}$ and $L_{2}$ are calculated as
(3)$$\begin{array}{l} L_{1}=norm(LV_{start}(i)-O_{LCS})\\ L_{2}=norm(LV_{end}(i)-O_{LCS}) \end{array}$$
where ${LV}_{start}$ is the start vector of the laser line, ${LV}_{end}$ is the end vector of the laser line, and $O_{LCS}$ is the origin of the laser. All vectors have values in the global coordinate system.

The ray/triangle intersection algorithm is updated to include the condition that if the calculated distance is between distances $L_{1}$ and $L_{2}$, then the intersection is within the working range of the sensor. If not, the algorithm returns a value of zero.

### 2.3. Preventing Laser Line Intersections with Overlapping Assembly CAD Parts

To simulate 2D/3D laser line sensor measurements in the Matlab 2023a, a CAD model needs to be imported. To import a CAD model into the Matlab, an STL format is needed, which can be generated in any modeling software such as SolidWorks 2023 or Catia v6. When solid CAD model is saved in STL format, it generates a lot of triangles on every surface of the CAD model. This means that triangles are on every side of the CAD model and that the laser vector line intersects with both, so the result is double distance, while only one result would be measured with the real sensor. The problem is shown in Figure 5, where laser lines are reduced for easier visualization of the problem.

The above-mentioned problem can be solved by checking the angle between the normal vector of the triangles and the normal vector of the laser coordinate system in the direction of the *z*-axis. If the angle between the normals is less than 0, then the normals point in different directions. This represents the upper triangles facing the laser coordinate sensor. If the assembly CAD model has more overlapping parts, another scenario could happen, which is shown in Figure 6.

In the case shown above, one laser line can intersect with four triangles. Of these four triangles, the lower triangle from the 2nd CAD model and the triangle facing the right side of the 1st CAD model generate an angle with the *z*-axis of the laser coordinate system that is greater than or equal to 0. These two triangles are eliminated by the condition already stated above. The remaining two intersection triangles generate an angle with the *z*-axis of the laser coordinate system that is less than 0, so the above condition does not eliminate the problem of overlapping.

The problem of the overlapping CAD models is eliminated with the help of a matrix with positive flags (ones), where the rows represent all triangles of the STL model, and the columns represent laser lines in one laser scan. The number of triangles can vary depending on the CAD model and the number of triangles oriented towards the *z*-axis of the laser coordinate system. If a laser line intersects with a triangle, 1 is set at the intersection of a laser line number and the number of a triangle in the flag matrix. The flag matrix can be seen in Table 2, where laser line 3 intersects with triangles 1 and 4. If one laser line intersects with two triangles or more, the smallest distance from the laser origin to the closest triangle needs to be taken into consideration.

To find the smallest distance from the laser origin to the closest triangle where overlapping occurs, a matrix containing the distances of each laser line at points of intersection with the triangles is required, as shown in Table 3.

Laser line number 3 overlaps with triangles 1 and 4. In the distance matrix, the smallest distance for laser line 3 is with triangle 1. Using this logic, the overlap of the laser line with multiple triangles is resolved.

### 2.4. Optimizing the Laser Line Measurements Algorithm

The above-mentioned algorithm checks all laser lines from each scan with all triangles that are facing in the direction of the laser origin. The assembly CAD model shown in Figure 1 generated in STL format has 3638 triangles. There are 1988 triangles that are facing in the direction toward the laser origin. In addition, 1988 triangles need to be checked to see if the intersection occurs with an individual laser line. One laser scan generates 2048 laser lines. If 1 mm resolution is needed, that would generate 450 scans. This would create a flag and distance matrix with a size of 2048 × 1988, which would need to be recalculated 450 times and would take a lot of processing power and computing time to generate simulated measurements.

To reduce the computing time for calculating simulated measurements, the virtual plane crossing the origin of the laser coordinate system is generated, as can be seen in Figure 7.

The principle of optimization is to calculate the perpendicular distance of each vertex point of triangles facing the origin of the laser to the generated plane. The following formula is used, where n is the normal vector of the plane, $V_{i}$ is the 3D coordinates of the vertices of the current triangle, and d is the distance from the origin to the plane.
(4)$$\begin{array}{l} D_{i}=n\cdot V_{i}-d\\ i=1-3 \end{array}$$

If the distances $D_{i}$ all have the same sign for one triangle, then the triangle is on one side of the plane. If the distances $D_{i}$ for one triangle have different signs, that means that the plane intersects with the triangle. The intersected triangle is flagged by the algorithm. Only flagged triangles are later checked if the generated laser lines intersect with specific triangles. With implementation of the above-mentioned principle, the number of triangle candidates in the i-th laser scan is reduced to 91. This number of course varies, and it depends on the current position and orientation of the laser coordinate system as well as number of triangles in the intersecting plane.

### 2.5. Including Laser Resolution into Simulated Measurements

To completely simulate measurements of the real 2D/3D laser line sensor, the resolution of the real sensor needs to be taken into consideration. The resolution of the MLWL 132 Wenglor sensor can be seen in Table 1. Since resolution is defined over the working area, it is the smallest when the measurement process is closer to the upper working range and the biggest when the measurement process is at the lower working range of the sensor. Since distance is important for defining the resolution, the simulated measurements on the *z*-axis are scaled according to the working range on the *z*-axis of the sensor.
(5)$$SF=\frac{Z_{meas}-WR_{z}(1,1)}{WR_{z}(1,2)-WR_{z}(1,1)}$$

In the above equation, SF is the scaling factor, $Z_{meas}$ is the measured distance on the *z*-axis of the laser coordinate system, and ${WR}_{z}$ is the working range array on the *z*-axis. The maximal value of the resolution on the *z*- and *x*-axes can be calculated as follows:(6)$$\begin{array}{l} {\mathrm{MaxRes}}_{z}=R_{z}(1,1)+\left(R_{z}(1,2)-R_{z}(1,1)\right)\cdot SF\\ {\mathrm{MaxRes}}_{x}=R_{x}(1,1)+\left(R_{x}(1,2)-R_{x}(1,1)\right)\cdot SF \end{array}$$
where ${MaxRes}_{z}$ and ${MaxRes}_{x}$ represent the maximal resolution possible for the current distance on *z*-axis for z and x measurements, and $R_{z}$ and $R_{x}$ represents the resolution array for the *z*- and *x*-axes. The resolution in the simulated measurements is then computed as follows:(7)$$\begin{array}{l} {\mathrm{Resolution}}_{z}=\frac{{\mathrm{MaxRes}}_{z}}{2}\cdot rand\left(\left[-100,100\right]\right)\cdot 0.01\\ {\mathrm{Resolution}}_{x}=\frac{{\mathrm{MaxRes}}_{x}}{2}\cdot rand\left(\left[-100,100\right]\right)\cdot 0.01 \end{array}$$
where ${Resolution}_{z}$ and ${Resolution}_{x}$ are computed according to the distance of the measurement on the *z*-axis within the resolution provided in datasheets. The calculated resolution is then added to the simulated measurements.

### 2.6. The Developed Algorithm Flowchart

To start simulation with the developed algorithm, an STL CAD model needs to be imported. Then, laser characteristics need to be defined as well as the laser initial position and orientation in GCS. After that, the laser trajectory over the imported STL CAD model needs to be defined, along with all desired outputs such as laser resolution, plotting laser working area, generating a point cloud in LCS and/or in GCS, as well as saving all the data. The flowchart of the developed algorithm can be seen in Figure 8.

### 2.7. Experimental Setup with a Real 2D/3D Laser Line Sensor

To compare simulated measurements of the 2D/3D laser line sensor with the real measurements, a linear unit with a servo motor was set up as can be seen in Figure 9. To drive the linear unit, a Siemens PLC S7-1500 was selected with an S120 CU310 control unit for the Siemens servo motor. The PLC communicates with the Matlab on the PC over TCP/IP protocol where Matlab is set up as the server. The MLWL 132 Wenglor sensor communicates with the Matlab over TCP/IP protocol where the Wenglor sensor is set up as the server. Each scan on the sensor is triggered over the DO of the PLC. The resolution of the scans is set to 1 mm, while the speed of the scanning is set to 5 mm/s on the PLC. The measuring results are sent over TCP/IP to the Matlab program, where they are saved in PLY format and in a CSV file.

## 3. Results

### 3.1. Simulated Measurements of CAD Models

The initial simulations of 2D/3D laser line measurements were performed on simple 3D models, which can be seen in Figure 10, on which the region of interest is marked as well as the direction of scans. For the L-part, V-part, and S-part, there are no real parts to compare the results. However, simulated measurements of a CAD model of assembly part are compared with the real measurements of the real part from the MLWL 132 Wenglor sensor. The simulations were performed on a computer with Intel(R) Core(TM) i7-9850H CPU @ 2.60 GHz.

Part dimensions, number of triangles, number of scans, and time needed to calculate simulated measurements with optimized and non-optimized algorithms are seen in Table 4.

As can be seen from the results, the time needed to perform simulated measurements increases with the complexity of the CAD model. The more triangles the CAD model contains, the more time is needed to calculate simulated measurements. The results of the simulated measurements can be seen in Figure 11, where for each part, the point cloud is shown in the laser coordinate system and in the global coordinate system.

### 3.2. Comparing Simulated Measurements and Real Measurements

The assembly CAD model part has been compared with the real model to validate the simulated measurements. The comparison has been made with the help of Matlab functions for point cloud manipulation using plane-to-plane registration with the Generalized Iterative Closest Point (G-ICP) method [14]. The results can be seen in Figure 12. The mean distance between simulated and measured point cloud data has been calculated with the help of the Matlab function “findNearestNeighbors”, from which the mean distance and maximal distance were calculated. The mean distance between simulated and measured point cloud data is 0.42 mm, while the maximal distance is 3.28 mm. The maximal distance is also shown in Figure 12c.

### 3.3. Resolution Simulation Results

The generated resolution of measurements in the simulation, compared with the resolution of measurements from the real sensor, is shown in Figure 13. It can be observed that the resolution of the simulated measurements behaves similarly to the resolution of the real sensor measurements.

### 3.4. Algorithm Deficiency

During simulating 2D/3D laser line measurements with the developed algorithm, one shortcoming of using the algorithm and STL format has been noticed. It appears that to successfully generate an STL CAD model, small triangles need to be generated, and it is possible for the laser line to be parallel to the triangle with zero distance but no intersection. The described problem is easily seen in Figure 14.

## 4. Discussion

To successfully simulate measurements of a 2D/3D laser line scanner, a modified ray/triangle method was used, incorporating the working area of the actual sensor. To speed up the simulation, only triangles facing the *z*-axis of the laser coordinate system and intersecting with the xz plane generated by the laser coordinate system were considered. However, these triangles might belong to different parts in an assembly CAD model, allowing the simulated laser line to intersect multiple triangles. This could result in calculating double distances, whereas in real measurements, the sensor would record only a single measurement. This issue was addressed using flags and distance matrices. If multiple elements in a row of the flag matrix—representing a simulated laser line—are equal to one, it indicates intersections with multiple triangles. These rows were then compared with the distance matrix to identify the minimal distance, representing the closest intersection to the laser origin.

The resolution of the real sensor was also included in the measurement simulation algorithm. The resolution of the real sensor depends on the working area of the sensor. The closer to the sensor origin, the smaller the resolution. As can be seen, the resolution of the real sensor was successfully integrated into the measurement simulation algorithm.

When comparing simulated measurements and real measurements from the sensor, some differences occur. It is impossible to find the exact trajectory of the simulated laser and real laser on a linear unit. It is also very hard to scan the same area in simulation and in the real part. Simulated scans of the assembly CAD model were parallel to the cross section of the model in the direction of the scan, while the scans of the real part with real sensors were approximately parallel to the cross section of the part. This misalignment can be seen in Figure 12c and Figure 13, where green lines (simulation) and violet lines (measurements) are not completely aligned.

During the simulation of measurements on the assembly CAD part, a deficiency of the developed algorithm was found. The generated point cloud from simulated measurements had missing points in a specific area. It has been found out that simulated laser lines were parallel to triangles with zero distance, so there were no intersections, as can be seen in Figure 14. This problem can be omitted if the initial position of the simulated laser trajectory is offset by a small distance, for example 0.1 mm.

## 5. Conclusions

The developed algorithm for simulating measurements of a 2D/3D laser line sensor successfully simulates measurements and creates point cloud data where the real laser resolution is also included. Simulated measurements are generated in a laser coordinate system and/or in a global coordinate system, depending on user input. The trajectory of the simulated laser can also be saved if needed. The developed algorithm can be of great help in generating data of point cloud measurements if a real sensor is not available or if a real part is still not made. It could also be used for data generation for training AI algorithms on different CAD parts where real parts do not have to be made. In this way, the developed algorithm can reduce the waste that could potentially occur by making a real model and reducing the carbon footprint.

In future work, studies will be performed on how to reduce the simulation time needed to simulate measurements. For more successful data training with AI algorithms, measuring noise could also be added in future work.

The developed algorithm is available within the Appendix A.

## Figures and Tables

**Figure 1 sensors-24-07396-f001:**
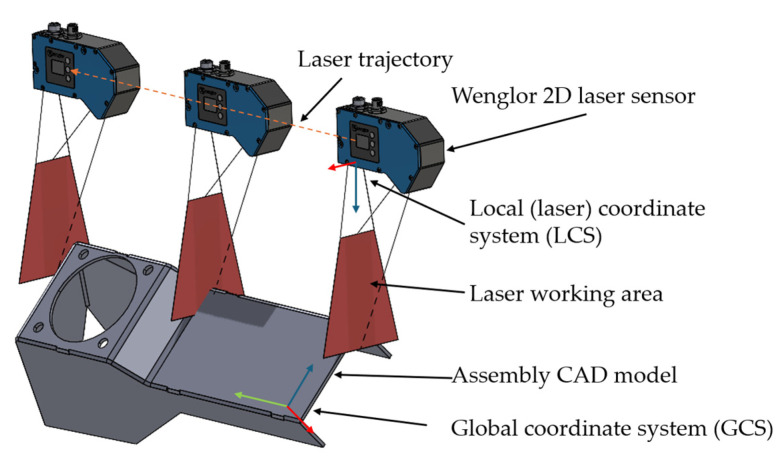
Simulation principle with the assembly CAD model and a 2D laser line sensor.

**Figure 2 sensors-24-07396-f002:**
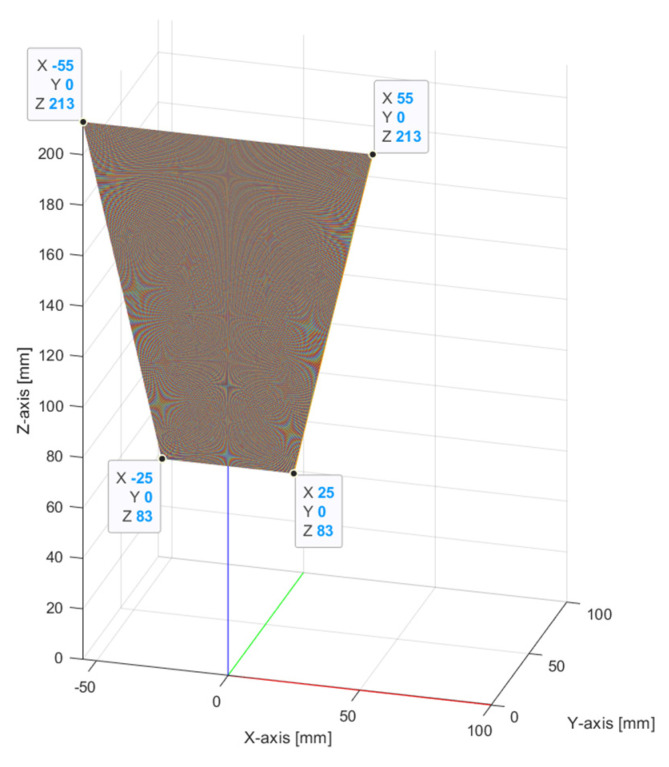
Generated laser lines with vectors within the working area.

**Figure 3 sensors-24-07396-f003:**
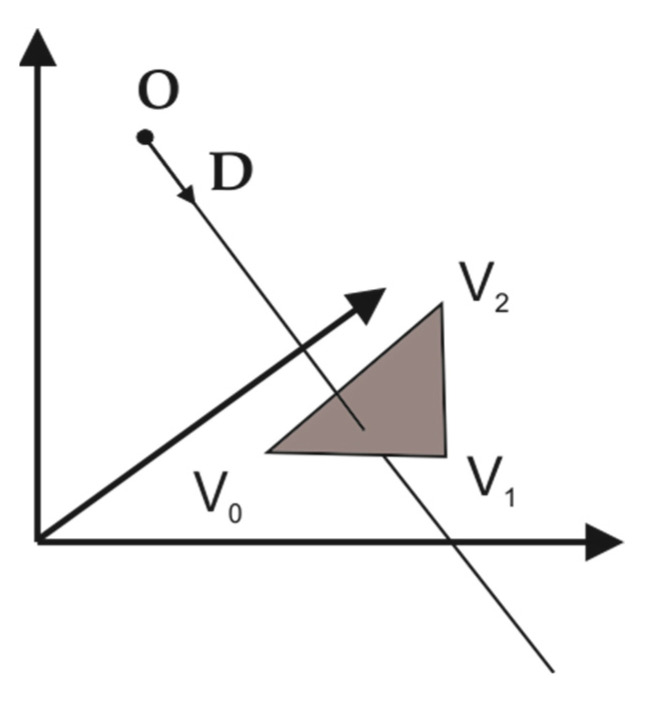
Ray/triangle intersection principle.

**Figure 4 sensors-24-07396-f004:**
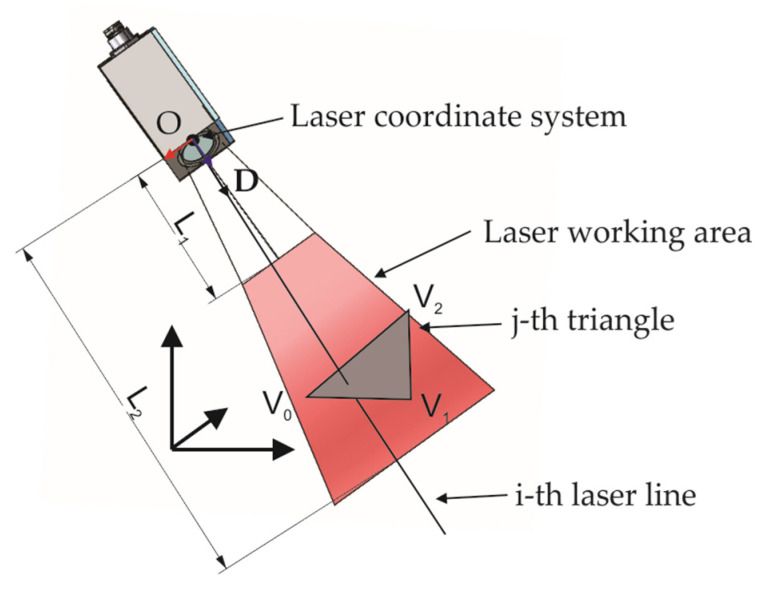
Modified ray/triangle intersection method.

**Figure 5 sensors-24-07396-f005:**
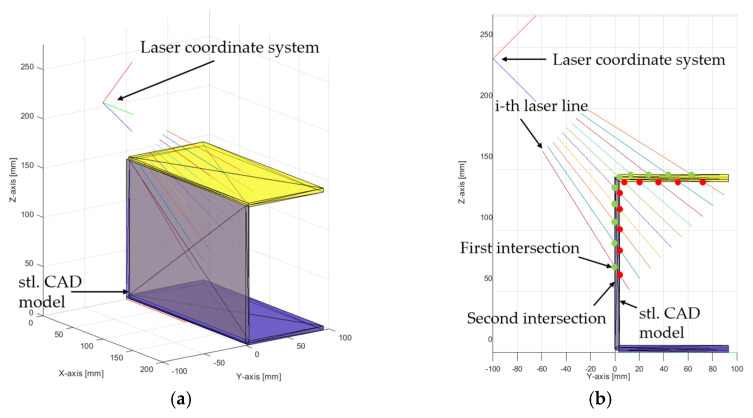
(**a**) STL CAD model with laser lines and (**b**) first and second intersection points of the i-th laser line.

**Figure 6 sensors-24-07396-f006:**
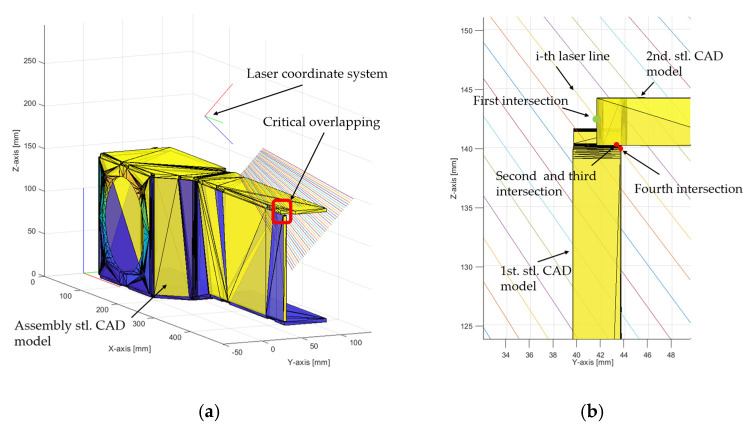
(**a**) Assembly STL CAD model with critical overlapping area and (**b**) detailed overlapping area with a laser line intersecting four triangles.

**Figure 7 sensors-24-07396-f007:**
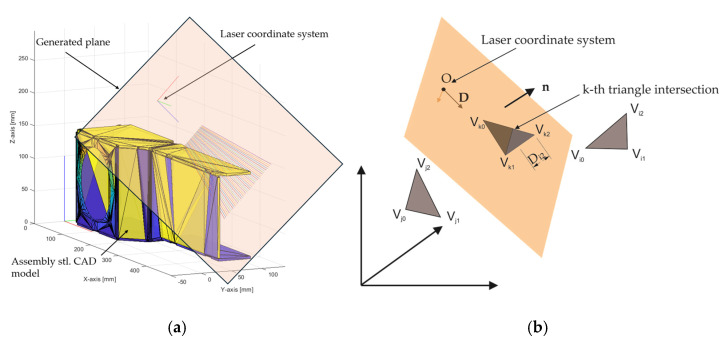
(**a**) Generated plane and the CAD model and (**b**) principle for finding intersecting triangles with the plane.

**Figure 8 sensors-24-07396-f008:**
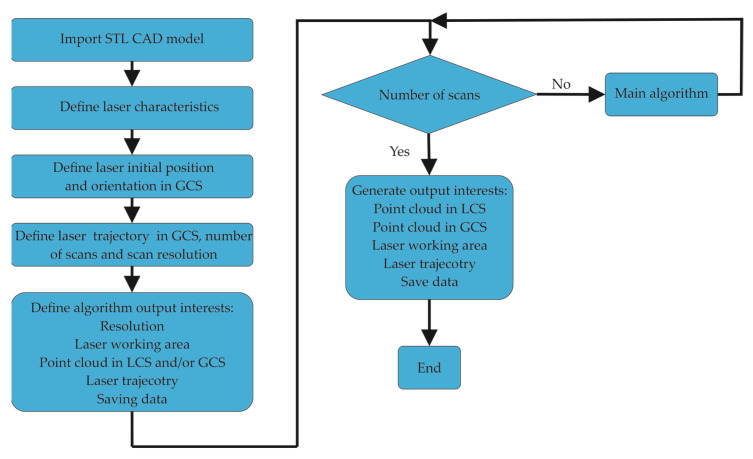
Flowchart of the developed algorithm.

**Figure 9 sensors-24-07396-f009:**
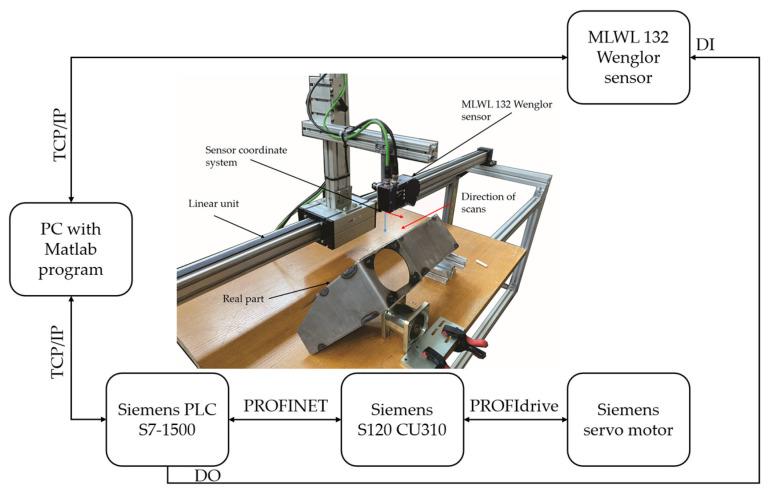
Experimental setup with a real Wenglor sensor, linear unit, and Matlab program.

**Figure 10 sensors-24-07396-f010:**
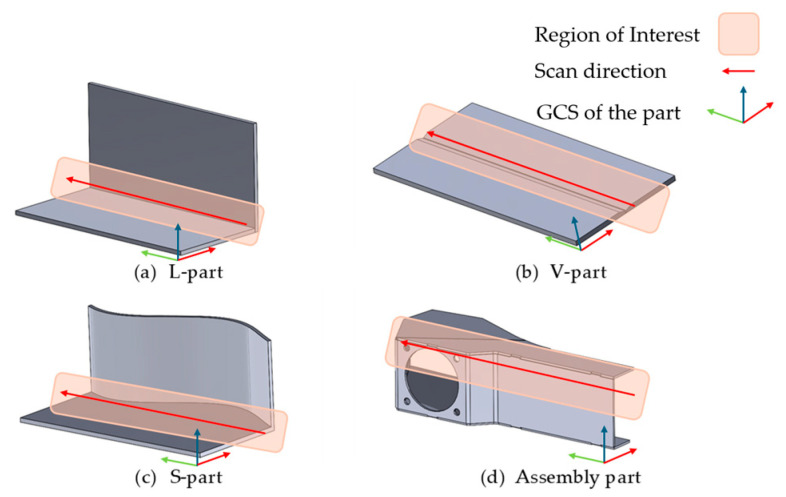
Parts on which laser measurements have been simulated and their regions of interest.

**Figure 11 sensors-24-07396-f011:**
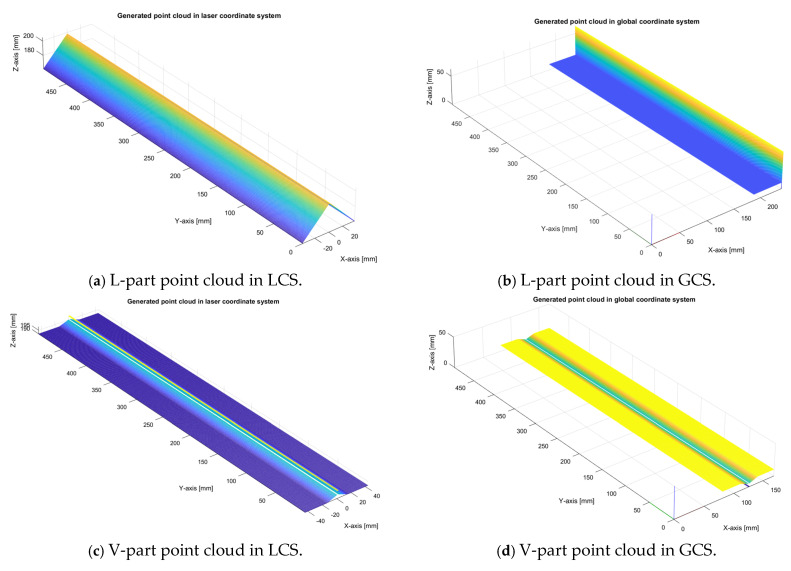
Generated point cloud results of each part, where (**a**,**c**,**e**,**g**) are measured in the laser coordinate system, and (**b**,**d**,**f**,**h**) are measured in the global coordinate system.

**Figure 12 sensors-24-07396-f012:**
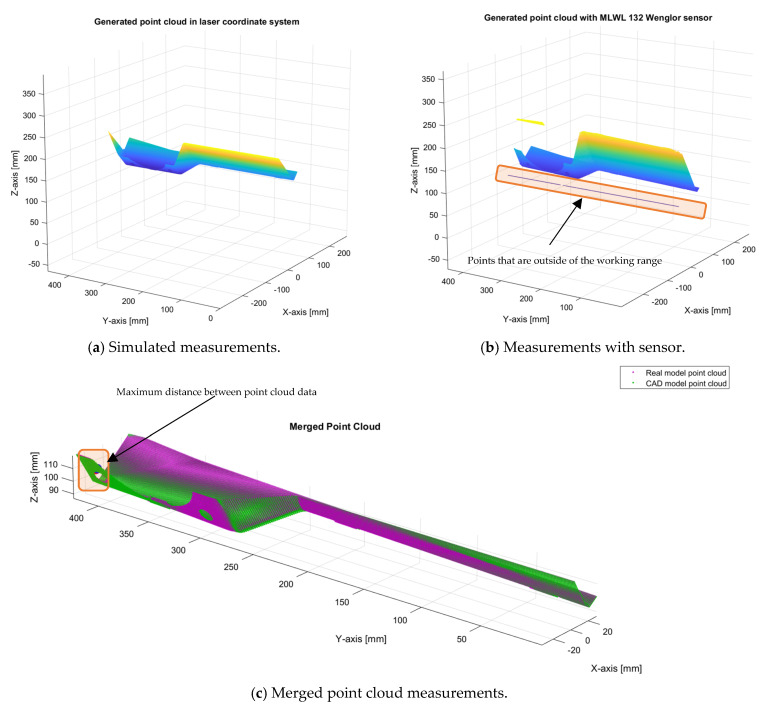
Comparing simulated point cloud measurements with point cloud measurements from a real sensor.

**Figure 13 sensors-24-07396-f013:**
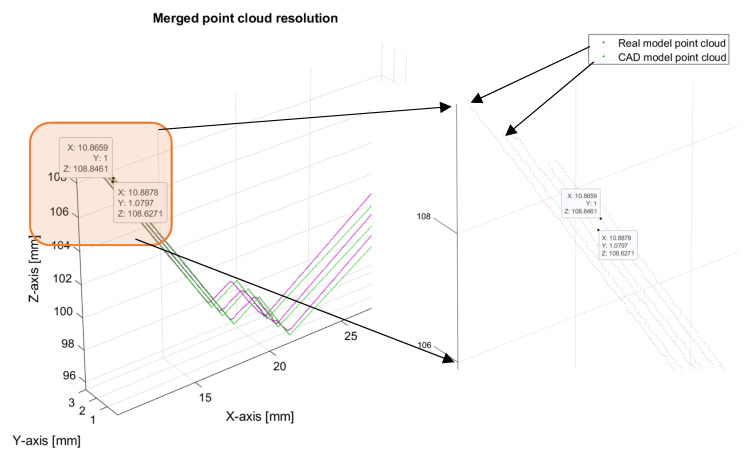
Comparing simulated resolution of measurements to sensor resolution of measurements.

**Figure 14 sensors-24-07396-f014:**
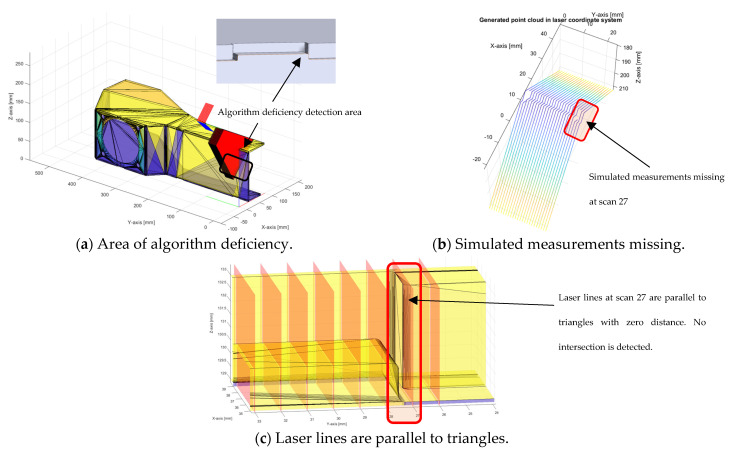
Detected algorithm deficiency at specific area of the assembly CAD STL model.

**Table 1 sensors-24-07396-t001:** Relevant sensor specifications of the 2D/3D MLWL 132 laser sensor.

2D/3D MLWL 132	
Working range Z	83–213 mm
Measuring range Z	130 mm
Measuring range X	50–110 mm
Resolution Z	3.2–14 μm
Resolution X	26–55 μm
Num. of points per scan	2048

**Table 2 sensors-24-07396-t002:** Flag matrix representation of laser line intersection with triangles.

Laser Lines\Triangles	1	2	3	4	5	6	7	8	9	…	i-th
1	0	0	0	0	0	0	0	0	1	…	0
2	0	0	0	0	0	0	0	0	1	…	0
**3**	**1**	**0**	**0**	**1**	**0**	**0**	**0**	**0**	**0**	**…**	**0**
4	1	0	0	0	0	0	0	0	0	…	0
5	0	1	0	0	0	0	0	0	0	…	0
6	0	1	0	0	0	0	0	0	0	…	0
⋮	⋮	⋮	⋮	⋮	⋮	⋮	⋮	⋮	⋮	⋮	⋮
2048	0	0	0	0	0	0	0	0	0	1	0

**Table 3 sensors-24-07396-t003:** Distance matrix representing distances from the laser origin and intersection point on the i-th triangle.

Meas. of Lines\Triangles	1	2	3	4	5	6	7	8	9	…	i-th
1	0	0	0	0	0	0	0	0	140.01	…	0
2	0	0	0	0	0	0	0	0	139.8	…	0
**3**	**139.1**	**0**	**0**	**144.5**	**0**	**0**	**0**	**0**	**0**	**…**	**0**
4	138.5	0	0	0	0	0	0	0	0	…	0
5	0	137.5	0	0	0	0	0	0	0	…	0
6	0	136.8	0	0	0	0	0	0	0	…	0
⋮	⋮	⋮	⋮	⋮	⋮	⋮	⋮	⋮	⋮	⋮	⋮
2048	0	0	0	0	0	0	0	0	0	145.6	0

**Table 4 sensors-24-07396-t004:** Part characteristics, number of scans and simulation time needed to calculate measurements.

CAD Model [STL]	Dimensions [mm×mm×mm]	Number of Triangles	Number of Laser Lines Per Scan	Number of Scans	Resolution of Scans [mm]	$\mathrm{Simulation}\;\mathrm{Time}\;(\mathrm{Optimized})\;\left[s\right]$	$\mathrm{Simulation}\;\mathrm{Time}\;(\mathrm{Non}-\mathrm{Optimized})\;\left[s\right]$
L-part	500 × 250 × 250	24	2048	500	1	1349	2389
V-part	500 × 250 × 250	36	2048	500	1	1454	3279
S-part	500 × 250 × 250	260	2048	500	1	2608	11,586 *
Assembly	558 × 210 × 133	3638	2048	460	1	8619	135,815 *

* Estimated time needed to perform simulated measurements.

## Data Availability

Data are contained within the article.

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
