# Peer review of "Simulating and Verifying a 2D/3D Laser Line Sensor Measurement Algorithm on CAD Models and Real Objects"

_sensors, 2024, doi:10.3390/s24227396_

Round 1
Reviewer 1 Report
Comments and Suggestions for Authors
1. The manuscript mentions that the measurement time of the algorithm has been reduced by optimizing the simulation algorithm. However, there is no quantitative study of the efficiency of the algorithm in the experimental part. The authors should provide a comparison of the efficiency of the proposed algorithm.
2. In the experimental part, I have doubts about the representativeness of the experimental part chosen by the author. In the analysis of the experimental results, the modeling effects of features on parts, such as holes and grooves, are not discussed.
3. In addition, there is a big difference between the experimental results and the experimental data, which is not further explained by the author. Please supplement the discussion of the experimental results.
4. There are some problems with the quality of graphics in the manuscript, as shown in Figure 11 on page 12. Please optimize the quality of graphics to improve readability.
Author Response
Thank you very much for taking the time to review this manuscript. Please find the detailed responses below and the corresponding revisions with track changes in the re-submitted files.

Reviewer 2 Report
Comments and Suggestions for Authors
The paper presented an algorithm in Matlab to simulate the measurements of any 2D/3D laser line sensor on any STL CAD model. Simulations showed the results of the proposed method. Concerns:
1. Contributions should be described in Introduction for the core ideas of the proposed paper.
2. Some related references should be review.
3. Simulations should be compared with some related works.
4. Figures should re-organzied. Some figures are too large, e.g., Figure 14.
5. Limitations and future work should be discussed to provide a comprehensive presentation. For example, some point cloud analysis [1] and LiDAR data analysis [2] could improve the task.
[1] Part-Whole Relational Few-Shot 3D Point Cloud Semantic Segmentation, Computers, Materials and Continua, 2024.
[2] Part-Whole Relational Fusion Towards Multi-Modal Scene Understanding, 2024.
Comments on the Quality of English LanguageN/A
Author Response

(The authors gave the same response as above.)

Reviewer 3 Report
Comments and Suggestions for Authors
In the manuscript titled "Simulating and verifying 2D/3D laser line sensor measurement algorithm on CAD models and real objects", Belšak et al. proposed a novel algorithm on Matlab that employs a modified ray/triangle intersection method for measuring arbitrary STL CAD models generated. Four different 3D models were measured, and their errors were found to be small when compared with actual measurement results. This method provides a reference for reducing training costs in artificial intelligence and also offers a reference for reducing waste generated by the production of physical models. However, some issues must be addressed before considering publication.
Comments
1. In the abstract on the first page, I did not see the purpose of developing this method. It is recommended to introduce the problem this method was developed to solve after presenting the general background.
2. On page 7, section 2.3, the authors write, "Flag matrix can be seen in Table 2 where laser line 3 intersects with triangles 1 and 4". Please explain how the triangle numbers are arranged to make this text, as well as Table 2 and Table 3, more comprehensible.
3. On page 7, in Table 2, why is the triangle number “…” intersecting with the 2048th laser line a number “1”? Please explain the meaning of the number “1”, as well as Table 3.
4. On page 13, Figure 12c), the point cloud at the edge seems to be mismatched. The authors claim that the maximum error comes from the distance of the boundary points of the two point clouds; however, the error should come from the region where the point clouds match. Please explain the rationality of this approach.
5. On page 14, Figure 13, to my knowledge, "simulated resolution" and "measured resolution" are not proper nouns, but I did not receive information about the resolution in the figure. Please clarify what the resolution represented in the figure is.
6. Many figures in the article are blurry, such as Figures 11 and 12, with faint and indistinct axes.
Author Response

(The authors gave the same response as above.)

Round 2
Reviewer 2 Report
Comments and Suggestions for Authors
1. The main novelty is marginal. For a publication, some deep insight into the filed should be provided instead of some experimental analysis.
2. How to prove the superiority of the proposed work? Some state-of-the-art methods should be taken for comparison in performance.
3. For a research paper, the reference are so few.
Comments on the Quality of English LanguageN/A
Author Response
We would like to point out that Reviewer 2 has downgraded ratings for our manuscript as can be seen in the above pictures. We would also like to point out that to our knowledge all the comments on the first round have been answered. Reviewer 2 has also suggested that two research papers were included in the manuscript. Since research papers had similar names, we could guess that reviewer 2 was one of them. We have decided not to include them. We found that the topics of the suggested research papers were not suitable for our research subject. We hope that our decision not to include the suggested research papers did not lead to downgrading of our manuscript.

Reviewer 3 Report
Comments and Suggestions for Authors
The revised manuscript "Simulating and Verifying 2D/3D Laser Line Sensor Measurement Algorithm on CAD Models and Real Objects" has been significantly improved after taking into account all the comments. Now it is recommended for acceptance.
Author Response
Dear Reviewer,
thank you for your valuable comments and time.
Best Regards,
Timi Karner, corresponding author